# Obesity, Fruit and Vegetable Intake, and Physical Activity Patterns in Austrian Farmers Compared to the General Population

**DOI:** 10.3390/ijerph19159194

**Published:** 2022-07-27

**Authors:** Sandra Haider, Maria Wakolbinger, Anita Rieder, Eva Winzer

**Affiliations:** Center for Public Health, Department of Social and Preventive Medicine, Medical University of Vienna, Kinderspitalgasse 15/1, 1090 Vienna, Austria; sandra.a.haider@meduniwien.ac.at (S.H.); anita.rieder@meduniwien.ac.at (A.R.); eva.winzer@meduniwien.ac.at (E.W.)

**Keywords:** physical activity, fruit and vegetable intake, alcohol, obesity, farmers

## Abstract

Low fruit and vegetable (F&V) intake, sedentary behavior, excessive alcohol consumption, and smoking are risk factors for the development of non-communicable diseases. This study describes the patterns and factors of nutrition (F&V and alcohol intake), physical activity (PA), obesity, and other chronic diseases of 10,053 adult farmers (52.7% female) in Austria, based on the cross-sectional survey from the Austrian Social Insurance Institution for the Self-Employed and compared with the results of the general Austrian population from 2019 (*n* = 14,606; 53.7% female). Compared to the general Austrian population, farmers showed a higher prevalence of overweight and obesity (42.8% vs. 36.5%; 18.8% vs. 17.1%), as well as hypertension, hypercholesterolemia, and diabetes mellitus. Additionally, farmers ate less F&V (0 servings/day 39.7% vs. 14.0%; 1–4 servings/day 55.5% vs. 80.8%) and only 4.8% vs. 5.1% (*p* < 0.001) fulfilled the F&V recommendations. Lower participation in endurance training (38.3% vs. 52.1%) was found, whereas farmers did more strength training (64.1% vs. 27.6%). Those who failed to fulfill the PA recommendations reported worse health status (OR: 3.14; 95%-CI: 2.08–4.76) and a higher chance for obesity (OR: 1.68; 95%-CI: 1.38–2.05). Since obesity rates among farmers are high and recommendations have rarely been met, every opportunity should be taken to promote healthy eating and adequate PA.

## 1. Introduction

Many chronic diseases are associated with risk factors such as unhealthy eating, excessive alcohol consumption, no regular physical activity (PA), and smoking [1]. By making healthy choices, the likelihood of developing non-communicable diseases, including obesity, can be reduced [2]. The onset of obesity results from a complex set of interactions between genetic, environmental, behavioral, and societal factors. Although many factors are involved in the development of obesity, the fundamental reason is an imbalance in energy intake and expenditure. There has been an increase in the intake of energy-dense products high in fat, sugar, and salt and a decrease in PA due to changes in global diets, food systems, accessibility of transportation, working conditions, and increased urbanization [3].

To maintain health and prevent non-communicable diseases, there are various recommendations for adults. For healthy eating, it is recommended to eat at least 1–2 servings of fruits and 3–4 servings of vegetables a day [4,5]. Concerning alcohol, the recommendation demands a maximum of two drinks for men and one drink a day for women [6]. For PA, the guidelines state that adults should exercise at least 150 min at moderate intensity or 75 min at a higher intensity/week or a combination of both [7,8,9], and strength training twice a week with the major muscle groups is demanded [7,8,9]. Additionally, quitting smoking is recommended [6].

Health outcomes of farmers differ in previously published studies. Farmers in Ireland and Crete (Greek) lack adherence to the fruit and vegetable (F&V) recommendation [10,11], whereas those in Taiwan achieved the fruit recommendation [12]. Generally in developing countries, small-scale family farming experiences problems in achieving food security [13]. In most studies, alcohol intake was reported to be high [14,15]. In Australia, people in farming communities are more likely to binge drink when compared with the general population [14]. In Brazil, the prevalence of alcohol abuse was 32% [15]. Concerning PA, most European studies in this population reported low PA [11,16]. The majority of studies in Europe and the USA reported high rates of obesity and overweight [11,17]. Moreover, Irish farmers have been identified to have higher rates of cardiovascular disease (CVD), compared to the general population [18], whereas, in Sweden, farmers had a lower risk of developing CVD, type 2 diabetes mellitus, and lower all-cause mortality compared to rural non-farmer and urban referents over 12 or 20 years [19,20].

This study aimed to compare the obesity prevalence and other chronic diseases, as well as F&V intake and PA patterns, in Austrian farmers with the general population. As a secondary objective of this analysis, we investigated whether farmers who did not meet the recommendations for PA and F&V intake had detrimental health and weight status.

## 2. Materials and Methods

### 2.1. Study Population

To compare the farmers with the Austrian general population, we used two different datasets. Firstly, survey data from the Austrian Social Insurance Institution for the Self-Employed (SVS), where all Austrian farmers are insured, were included. In 2019, 273,021 farmers were insured, of which 33,285 received the questionnaire by mail. This random sample was chosen by the SVS based on the person’s characteristics, which correspond to the general population of Austrian farmers. The exclusion criteria were (1) multiple insurances, (2) persons < 18 years, and (3) relatives. Finally, 10,426 returned the questionnaire, corresponding to a response rate of 31.3%. Secondly, data from the Austrian Health Interview Surveys (ATHIS) 2019 [21], the Austrian version of the European Health Interview Survey (EHIS) [22], were used. This survey was conducted by the Austrian National Statistics Agency (Statistik Austria) and imputed missing values based on sex, age, education, and region of residence. Between 2018 and 2019, the ATHIS was conducted via computer-assisted personal interviewing (CAPI) and a web-based questionnaire. In total, 15,461 persons ≥ 15 years were included (14,225 people from CAPI; 1236 people from the web-based questionnaire), corresponding to a response rate of 50.5% [21]. Since the first dataset included farmers ≥ 18 years, we excluded those in the age group 15–19 years in the ATHIS dataset (as it was assessed in 5-year-old age groups) and we excluded farmers for whom the age was not known.

### 2.2. Measurements

The following variables or questions were present in both datasets:

Health status was assessed in the farmers’ dataset by asking, “My state of health is (1) very good; (2) good; (3) moderate and (4) poor”. In the ATHIS, health status was given in the categories (1) very good; (2) good; (3) moderate; (4) poor and (5) very poor [23]. To have comparable results, categories 4 and 5 of the ATHIS were combined.

Chronic diseases were assessed in the farmers’ dataset by asking, “I have elevated blood pressure, cholesterol, or blood glucose levels”, with the response option yes or no. Thus, these are self-reported data. People, who did not provide information were assumed not to have one of these chronic diseases. In the ATHIS, the question “Have you had any of the following illnesses or health problems in the last twelve months?”, with 19 different answer options, was asked [23]. Among these were hypertension, hypercholesterolemia, and diabetes mellitus.

Body mass index (BMI): In both datasets, body weight and height were self-reported data [23]. BMI was calculated (kg/m^2^) and categorized into underweight (<18.5 kg/m^2^), normal weight (18.5–24.9 kg/m^2^), overweight (25.0–29.9 kg/m^2^), and obesity (≥30 kg/m^2^) with class I, II, and III [24].

Food intake (F&V and alcohol): In the farmers’ dataset, food intake was collected with a modified version of the dietary habits questions used in the EHIS [25], which are adapted from the validated Single Question for Fruit and Vegetable [26]. To assess fruit intake, the following statement was given: “I eat at least one serving of fruit. One serving receives, e.g., one handful of carrots; two hands of salad”. The same procedure was performed for vegetables (without potatoes). For alcohol intake, the same statements as above were used with the following portion sizes: small beer (0.3 L), 1/8 L glass of wine, and 2 glasses of liquor. For each question or statement, the following answers were possible: (1) never or seldom; (2) <1 x/week; (3) 1 x/week; (4) 2–3 x/week; (5) 4–6 x/week; (6) 1–2 x/day; (7) 3–4 x/day; (8) ≥5 x/day. Based on these results, servings/day were calculated by converting the quantities based on the study of Yaroch et al. (never or seldom = 0; <1 x/week = 0.0715; 1 x/week = 0.143; 2–3 x/week = 0.357; 4–6 x/week = 0.715; 1–2 x/day = 1.5; 3–4 x/day = 3.5; >5 x/day = 5) [27]. In the ATHIS, the question. “How often do you eat fruits?” with the option (1) daily or several times/day; (2) 4–6 x/week; (3) 1–3 x/week; (4) <1 x/week; (5) never, was asked. Further: “How many servings of fruits do you eat per day?” was asked [22,23]. The same was done for vegetables or salad. The servings/day were also calculated as in the farmer’s dataset, based on the study of Yaroch et al. [27]. Additionally, alcohol intake was asked about as follows: “How often have you drunk alcohol in the last twelve months?” (1) daily or almost every day for the last year; (2) 5–6 days/week; (3) 3–4 days/week; (4) 1–2 days/week; (5) 2–3 days/month; (6) once a month; (7) <once a month; (8) not in the last 12 months, because I no longer drink alcohol; (9) never or only a few sips in my life [22,23].

Comparable to the Eurostat, daily consumption was divided into 0 servings, 1–4 servings, and ≥5 servings [28]. To assess the fulfillment of the recommendations concerning F&V, a cut-off of ≥1–2 x/day and ≥3–4 x/day were used, respectively [4,5]. For alcohol, the recommendations of <2 drinks/day for men or <1 drink/day for women were used [6].

Physical activity (endurance & strength training): In the farmers’ dataset, PA was assessed comprising two questions: “In a typical week, I spend _ hours _ minutes in my free time on sports, fitness or PA (e.g., Nordic walking, running, gymnastics, cycling, etc.)—not counting the activity included in strength training.” Strength training was assessed by asking: “In a typical week, on how many days do you practice special exercises to strengthen the muscles?”. In the ATHIS dataset, PA was assessed with the EHIS-Physical Activity Questionnaire (EHIS-PAQ) [23,29]. The number of days and duration of at least ten minutes of sports, fitness, or other PA without a break were asked. Time spent cycling to get around was also added, and a question about the frequency of activities to build up strength [23]. According to the WHO guidelines [8], this item was categorized as endurance recommendations fulfilled (≥150 min/week) and not fulfilled (<150 min/week). Further, strength training was categorized as recommendation fulfilled (≥2 x/week) and not fulfilled (<2 x/week) [8].

Covariates: The variables age (in the ATHIS dataset in 5-year-old age groups), sex, and smoking status were asked for. To have comparable results in education attainment level, five groups were formed (1) compulsory school; (2) professional educational (including apprenticeship, craftsmen); (3) secondary school (including college); (4) university (including university courses, universities of applied sciences); and (5) others (including “Meister” in German, Level 6 of the European Qualifications Framework).

Furthermore, some variables were only available in the farmers’ dataset:

Treatment of chronic diseases: The used treatment was asked with multiple response options, (1) no treatment; (2) medication use; (3) lifestyle modification. If participants combined no treatment with medication, the answer was rated as invalid. If they had chosen no treatment with lifestyle modification, lifestyle modification was chosen.

Subjective body weight estimation was asked by the question: “I consider myself as (1) underweight; (2) normal weight; (3) overweight, trying to reduce weight; (4) overweight, not currently trying to reduce weight; (5) overweight, feeling comfortable with it.”

### 2.3. Statistics

Categorical data are shown in percentages, and differences were calculated using Chi-Square-Tests. Metric normally distributed data are shown in mean and standard deviations (SD). To compare groups, *t*-tests for independent samples were used. To assess the association between not fulfilling the PA recommendations with health status, a univariate logistic regression analysis, was calculated. In a further step, this regression analysis was adjusted for sex, age, and educational attainment level. The same was performed for BMI categories and subjective body weight estimation. In the same way, the associations between 0 servings of F&V/day with health status, BMI, and subjective body weight estimation were determined. The data were analyzed using IBM^®^ SPSS^®^ Statistics Version 25 software (IBM Corp., Armonk, NY, USA). All statistical significance was set at *p* < 0.05. *p* values were not adjusted for multiple tests and should be interpreted explanatorily only.

## 3. Results

10,426 farmers and 15,461 persons of the general Austrian population provided answers. To obtain comparable samples, we excluded 373 farmers due to missing age, leading to a final sample of 10,053. From the ATHIS dataset, we excluded 855 people aged 15–19 years, leading to a sample size of 14,606. Therefore, in total, 24,659 were included in this analysis.

The comparison between farmers and the general Austrian population is shown in Table 1. There was no significant difference in sex, the majority of the farmers had finished a professional education, and one-eighth had completed secondary school or attended university. Further, farmers were less likely to smoke; however, they rated their health status to be worse. Farmers were more likely to experience hypertension, hypercholesterolemia, and diabetes mellitus. As such, 17% of all farmers ≥60 years suffer from diabetes mellitus, whereas the highest prevalence (23%) was found in those between 80 and 85 years. Moreover, farmers had a 6% and 2% higher prevalence of overweight and obesity, and more were living with obesity class I compared to the general population.

Concerning food intake, farmers ate less servings of fruits/day (0.9 (SD: 0.9) vs. 1.1 (SD: 0.9) (*p* < 0.001)) and vegetables (0.08 (SD: 0.8), 1.0 (SD: 0.7); *p* < 0.001) in comparison to the general population. In other words, one in three farmers reported not consuming any F&V daily, compared to one in seven of the general population. Only 4.8% of the farmers as well as 5.0% of the general population consumed the recommended five servings or more per day (Figure 1). On average, above half of the farmers and 80% of the general population reported eating between one and four servings of F&V/day (Figure 1).

Fewer farmers met the recommendations for fruit and also for the combination of F&V, compared to the general population. The difference in the fulfillment of the recommendation for vegetables was not significant. However, the majority of farmers and the general population met the recommendations for alcohol consumption (Figure 2).

Concerning PA, farmers reported significantly less endurance training participation, and 38% met the recommendations of ≥150 min/week. For strength training, farmers were more likely to meet the recommendations than the general population. Overall, one in seven farmers reported performing endurance training ≥150 min/week and strength training ≥2 x/week, compared with one in five in the general population (Figure 3).

Reported data on the treatment of various conditions and the subjective body weight estimation are shown in Table 2. Hypertension in farmers was mainly treated with medications, whereas lifestyle modification played a minor role. According to the self-reported data, hypercholesterolemia often remained untreated and lifestyle intervention unnoticed. Farmers with diabetes mellitus or high blood glucose levels mainly used medications, followed by no treatment, and a combination of medication use and lifestyle modification. Between 2% and 10% of the participants reported lifestyle modification as being their choice of treatment. Concerning the subjective body weight estimation, although the prevalence of overweight and obesity was higher than in the general population, the majority of the farmers reported having a normal weight and one-third reported suffering from overweight and trying to reduce weight.

The logistic regression analysis with adjustments for sex, age, and educational attainment level, showed that farmers who failed to meet the PA recommendations have a 3.1-times higher chance of experiencing poor health status, 1.3-times of overweight, and 1.7-times of obesity. They rate themselves 3.1-times and 1.9-times more likely as overweight but not trying to reduce weight, or rather feeling comfortable with it (Table 3).

Concerning F&V intake, the adjusted logistic regression analysis showed that farmers who ate no servings of F&V daily had a 1.5-times higher chance of experiencing poor health status and were 1.5-times more likely to consider themselves overweight but not trying to reduce weight (Table 4).

## 4. Discussion

The present analysis shows that, compared to the general population in Austria, the prevalence of overweight, obesity, and chronic diseases was higher in farmers. Farmers were less likely to meet the recommendations on F&V and endurance training; however, the recommendations for alcohol consumption were met. Farmers were less likely to smoke, and more likely to perform the recommended strength training than the general population.

The high prevalence of obesity in farmers (19%) is very alarming when compared to the general population in Austria (17%) and in the European Union (17%) [30]. This higher prevalence is comparable to previously published studies [31]: for example, 39% in Kentucky, Tennessee, and West Virginia, USA (*n* = 100) [17], 35% in Ireland (*n* = 310) [32], 28% in South Carolina, USA (*n* = 1,394) [33], 16% in Spain (National Health Survey, *n* = 12,037) [34], or 13% in France (AGRIculture and CANcer cohort, *n* = 180,060) [35]. In this context, it has to be mentioned that present data on BMI are based on self-reports. A limiting factor of this survey method is that self-reports were shown to overestimate body height, whereas weight is underestimated [36,37,38,39]. This underestimation of body weight was more common in women [37,40], and in individuals with overweight [37,39]. This fact must necessarily be taken into account when interpreting the data.

Concerning non-communicable or chronic diseases, we found not only a high prevalence of obesity but also a higher percentage of hypertension, hypercholesterolemia, diabetes mellitus, and poorer health status compared to the general Austrian population. When comparing the present data, for example with farmers in China (*n* = 4040), the authors observed a prevalence of hypertension of 42%, with comparable data on overweight (43%) and obesity (13% male and 19% female) [41]. In Ireland, 87% of farmers reported their health as being ‘good’ or ‘very good’ which was comparable with the national average of Irish males [11]. In this sample, hypertension was stated by 40%, hypercholesterolemia by 46%, and diabetes mellitus by 24% [11]. For all three chronic diseases, we observed lower prevalence rates. The reason for this could be the lower value of overweight in our sample. Notably, in this study, it was stated that the majority of people with elevated levels did not take any medication [11]. This is comparable to our results, which showed that one-third of farmers with diabetes mellitus and hypertension reported no treatment. Lifestyle interventions also play a subordinate role. However, there are also studies describing a lower risk of chronic disease in the farming population. One of them is a longitudinal observational cohort study in Swedish farmers (*n* = 1220), measuring a low relative risk of developing type 2 diabetes and CVD in comparison to the non-farming rural referents. Over an observational period of 20 years, 8% of farmers established type 2 diabetes, whereas 12% of non-farming rural developed it. The low risk was explained by lifestyle factors; physical capacity and meal quality [19]. It must be mentioned here that the baseline data were collected in 1990 in Sweden and that the working conditions of farmers may have changed since then.

When comparing the F&V consumption to the general population in Europe (Eurostat data), the majority of adults (55%) ate one to four servings, 33% ate zero servings and 13% met the recommendations of above five servings/day [28]. These results were adjusted for age, sex, and educational attainment level. Consequently, in our population of farmers, we observed a lower F&V intake compared to the Austrian but also to the European general population. In this ranking of the European Union member states, Austria, together with Bulgaria and Slovenia, were in last place. However, the Eurostat data are not fully comparable with our dataset because of the different response options and also due to the different survey methods (computer-assisted personal interviewing vs. questionnaire by mail). Nonetheless, other studies have also shown that farmers have difficulty achieving the recommendations [10,11]. A recently published Irish study showed that 79% of farmers did not achieve the recommended five servings of F&V/day [11]. This is in line with the longitudinal results from Crete of lacking adherence to the Mediterranean diet, which increased the risk of developing CVD [10]. On the other hand, a Taiwanese study, assessing the risk of colon cancer showed that farmers had a lower risk of colon cancer which was attributed to higher consumption of vegetables [12]. The differences are likely to be due to the different socio-demographic situations as well as working conditions in the countries. Our data also show that people who eat more than zero servings of F&V have a significantly better health status, probably due to the known properties of F&V intake such as containing essential vitamins and minerals, antioxidants, fibers, and bioactive plant substances [42]. However, there was no correlation between the BMI categories and F&V intake.

Besides, our data reveal that the majority of respondents complied with the recommendation regarding alcohol. This rate is in line with the results of a survey conducted on Irish farmers with only 10% reported drinking more than 17 standard drinks/week within health checks [11]. However, many studies reported a high rate of alcohol intake [14,15]. Since our data were collected by the farmers’ own insurance company and the Irish data were based on personal interviews [11], underreporting may have occurred on alcohol intake. Additionally, alcohol consumption is based on self-reports, which might lead to a further bias. As such, it was mentioned in the literature that in population surveys, an underestimation of alcohol intake of approximately 40–50% occurs [43].

A few more words about physical activity. In general, farming has always been considered a physically active working condition. However, in recent times, with the increased use of technology and machinery, farmers may experience more sedentary working conditions than before with a decreased occupational PA. For example in Irish farmers, 67% reported a high level of occupational PA with at least 30 min on five days/week or more [11]. In another study, 29 farmers were equipped with accelerometers and showed 124 (SD: 43) minutes of moderate-vigorous intense PA daily and eight hours of sitting time [44]. Since accelerometers have to be worn all day long, occupational and leisure-time PA was recorded. However, as recently published, it is essential to distinguish between occupational and leisure-time PA. In that regard, leisure-time PA reduced the risk for major adverse cardiovascular events and all-cause mortality, while occupational PA increases it [45]. This is one of the reasons, why we did not assess occupational PA in our study. Our data revealed that farmers were less physically active in their leisure time in comparison to the general Austrian population and also in contrast to the adults of the European Union, where 32% performed ≥150 min/week [46]. A representative sample of farmers in Poland (*n* = 2039) showed that, comparably, 33% met the recommendations of leisure-time PA [16]. Surprisingly, farmers performed more strength training. The reasons for this can only be hypothesized and it would be interesting to investigate this further. Despite this, the results showed that farmers who met the PA recommendations reported better health status and a lower risk of overweight and obesity. Accordingly, this states the importance of implementing targeted interventions to achieve the PA recommendations.

The risk of developing non-communicable diseases might be reduced in the farming population by behavioral and structural prevention through, e.g., targeted and multi-component interventions. A survey revealed the importance of providing health behavior services to farmers to improve their overall health [47]. Health promotion programs were also offered in Austria by their insurance company. However, only a few farmers (11%) have taken advantage of this offer (data not shown). Nevertheless, these prevention activities are to be further promoted. Prevention tasks include measures to prevent overweight and obesity, as well as the further intensification of care for those already affected. To positively influence body weight, a combination of nutritional and PA interventions should be implemented [3,48]. Another possibility is to increase health knowledge, an essential mediator for promoting weight stabilization [49]. Initiatives raising awareness of the PA [7,8,9] and nutritional recommendations [4,5] should be further expanded. A pilot study tested a 6-week community-based PA and health education intervention among Irish farmers with improvements in anthropometric measures, cardiovascular fitness, blood lipids, or dietary intake [50]. However, improving weight and lifestyle patterns require a sustained public health effort, which addresses not only individual factors such as knowledge but also the environmental conditions in which people live [51]. Therefore, Kinnunen and colleagues [52], demanded initiatives from the insurance company to become aware of issues concerning safety and health at work. A study was conducted to examine the views of agricultural professionals regarding challenges and opportunities in workplace health and safety. Barriers included access to health care, education, cultural competency, discrimination, logistics, economic considerations, and the labor contract system. These findings showed that challenges and opportunities are interrelated and overlapping [53]. This emphasizes the importance of creating a health-promoting environment, especially for this specific population, exposed to various barriers.

A strength of this study is the large sample size and the random sample. Moreover, there is still little data on this population. The main limitation is that all data are based on self-reports. However, self-reported BMI was shown to be a valid measure across different socio-demographic groups [54], and the used EHIS-PAQ showed reliability and validity for assessing PA [55]. Another limitation is that we have compared self-reported data to interview-assisted data. It should also be noted that the farmers’ data was collected by their insurance company, which could lead to a bias, as respondents may have given desired answers. Additionally, only F&V intake was inquired as an indicator of healthy eating due to the feasibility, and no comprehensive picture of the diet can be given. Noteworthy, there were minor differences in the two used datasets (e.g., health status was surveyed at a different numerical level, and F&V intake was assessed differently). However, this has been described extensively in the methods and has been calculated accordingly to be comparable. The cross-sectional design of the study is another limitation, as no causality can be drawn. Since data from 2009 are also available from Austrian farmers, the time course of nutritional, PA, and BMI development should be investigated in future analyses.

## 5. Conclusions

As obesity rates were high among farmers and recommendations on PA and F&V consumption were rarely achieved, every opportunity should be taken to promote healthy eating and adequate PA. Therefore, opportunities should be created to achieve the recommendations through behavioral interventions and structural prevention in the context of targeted and multi-component interventions.

## Figures and Tables

**Figure 1 ijerph-19-09194-f001:**
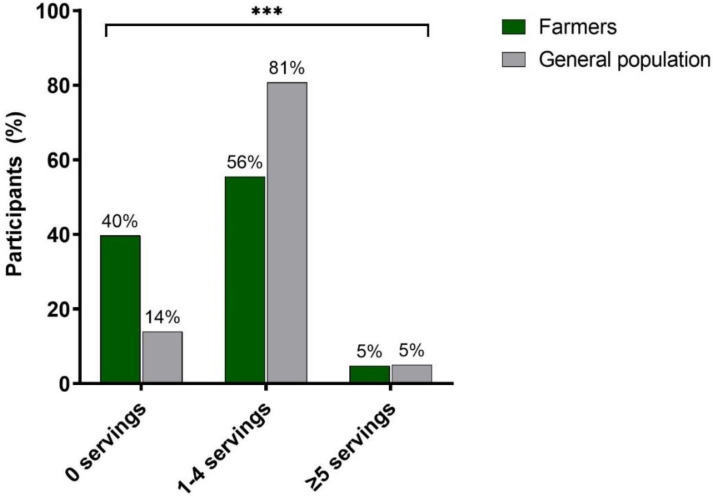
Participants’ percentage of daily intake of fruit & vegetables. Notes: *** significant differences between groups, which were calculated using Chi-Square Test.

**Figure 2 ijerph-19-09194-f002:**
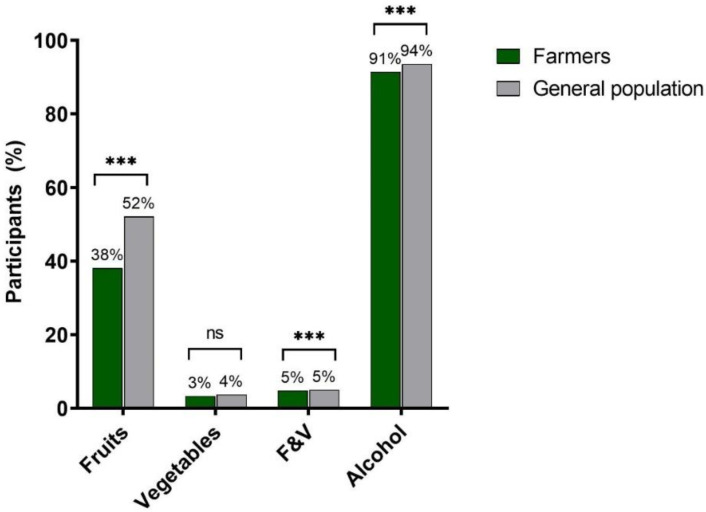
Participants’ percentage of recommendations’ fulfillment for fruit, vegetables, and alcohol. Notes: n.s. not significant; *** significant differences, which were calculated using Chi-Square Tests. To assess the fulfillment of the recommendations concerning F&V, cut-offs of ≥1–2 x/day and ≥3–4 x/day were used, respectively [4,5]. For alcohol, the recommendations of <2 drinks/day for men or <1 drink/day for women were used [6].

**Figure 3 ijerph-19-09194-f003:**
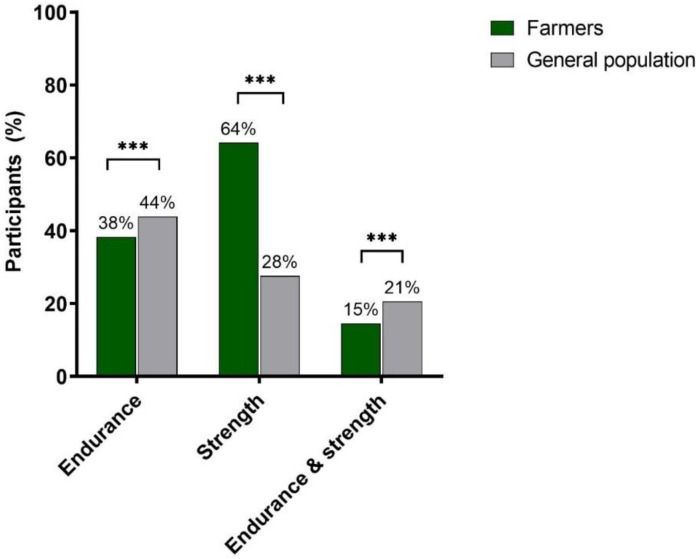
Participants’ percentage of recommendations’ fulfillment for physical activity. Notes: *** significant differences, which were calculated using Chi-Square Tests. Endurance recommendations were categorized as fulfilled (≥150 min/week) and not fulfilled (<150 min/week). Strength training was divided into fulfilled (≥2 x/week) and not fulfilled (<2 x/week) [8].

**Table 1 ijerph-19-09194-t001:** Characteristics of farmers in comparison to the general population.

	Farmers	General Population ^†^	
	*n* = 10,053	*n* = 14,606	*p*-Value
**Sex**, female	52.7%	53.7%	0.130
**Educational attainment**			
Compulsory school	27.0%	17.5%	<0.001
Professional education	39.1%	53.1%	
Secondary school	5.8%	14.7%	
University	3.6%	14.6%	
Others	24.5%	0%	
**Smoking**			
Every day	6.0%	19.1%	<0.001
Occasionally	2.9%	4.8%	
Not anymore, no	91.1%	76.1%	
**Health status**, score	2.3 (0.8)	2.0 (0.9)	<0.001
**Hypertension**	36.8%	24.6%	<0.001
**Hypercholesterolemia**	24.4%	20.1%	<0.001
**Diabetes mellitus**	10.8%	6.6%	<0.001
**BMI**, kg/m^2^	26.6 (4.4)	26.0 (4.7)	<0.001
Underweight (<18 kg/m^2^)	1.0%	1.9%	<0.001
Normal weight (18.5–24.9 kg/m^2^)	37.4%	44.5%	
Overweight (25.0–29.9 kg/m^2^)	42.8%	36.5%	
Obesity (≥30 kg/m^2^)	18.8%	17.1 %	
Class I (30.0–34.9 kg/m^2^)	14.6%	12.7%	<0.001
Class II (35.0–39.9 kg/m^2^)	3.1%	3.2%	
Class III (≥40 kg/m^2^)	1.1%	1.2%	

Data are given in mean and standard deviations or percentages. Differences in groups were calculated using Chi-Square-Test in categorical data, and *t*-Tests for independent samples. ^†^ unweighted data of the ATHIS.

**Table 2 ijerph-19-09194-t002:** Reported treatment of various conditions and subjective body weight estimation.

**Treatment of Hypertension (*n* = 3595)**	
None	10.8%
Medication use	81.0%
Lifestyle modification	2.4%
Combination of medication & lifestyle modification	5.3%
No valid answer	0.5%
**Treatment of hypercholesterolemia (*n* = 2253)**	
None	33.2%
Medication use	49.0%
Lifestyle modification	10.3%
Combination of medication & lifestyle modification	5.7%
No valid answer	1.8%
**Treatment of diabetes mellitus (*n* = 1028)**	
None	35.2%
Medication use	41.1%
Lifestyle modification	7.5%
Combination of medication & lifestyle modification	15.4%
No valid answer	0.9%
**Subjective body weight estimation**	
Underweight	2.3%
Normal weight	58.7%
Overweight, trying to reduce weight	29.6%
Overweight, not currently trying to reduce weight	3.3%
Overweight, feeling comfortable with it	6.0%

**Table 3 ijerph-19-09194-t003:** Association between not fulfilling the physical activity recommendations and health status, BMI categories, and subjective body weight estimation of farmers.

Physical Activity Recommendations Not Fulfilled							
		Crude Model			Adjusted for Sex, Age, Educational Attainment Level		
	%	OR	95% CI	*p*-Value	OR	95% CI	*p*-Value
**Health status**							
Very good	79.6%	1			1		
Good	86.1%	1.60	1.35–1.89	<0.001	1.77	1.48–2.12	<0.001
Average	87.1%	1.73	1.44–2.08	<0.001	2.13	1.73–2.62	<0.001
Poor	90.3%	2.38	1.60–3.54	<0.001	3.14	2.08–4.76	<0.001
**BMI categories**							
Normal weight	83.1%	1			1		
Overweight	85.7%	1.22	1.07–1.40	0.004	1.25	1.08–1.44	0.002
Obesity	89.1%	1.67	1.37–2.02	<0.001	1.68	1.38–2.05	<0.001
**Subjective body weight estimation**							
Normal weight	83.6%	1			1		
Overweight, trying to reduce weight	87.2%	1.34	1.16–1.55	<0.001	1.32	1.14–1.53	0.020
Overweight, not trying to reduce weight	94.4%	3.34	1.90–5.97	<0.001	3.14	1.79–5.54	0.001
Overweight, feeling comfortable	90.5%	1.86	1.33–2.60	<0.001	1.93	1.37–2.73	<0.001

Results are based on logistic regression analysis. Data are presented as odds ratios (OR) with a 95% confidence interval (95% CI).

**Table 4 ijerph-19-09194-t004:** Association between no servings of fruit and vegetables per day and health status, BMI categories, and subjective body weight estimation of farmers.

No Servings of Fruit and Vegetables Per Day							
		Crude Model			Adjusted for Sex, Age, Educational Attainment Level		
	%	OR	95% CI	*p*-Value	OR	95% CI	*p*-Value
**Health status**							
Very good	33.4%	1			1		
Good	39.7%	1.31	1.14–1.51	<0.001	1.31	1.12–1.52	<0.001
Average	42.2%	1.45	1.26–1.68	<0.001	1.41	1.20–1.67	<0.001
Poor	43.3%	1.52	1.21–1.91	<0.001	1.47	1.14–1.88	0.002
**BMI categories**							
Normal weight	37.7%	1			1		
Overweight	41.7%	1.18	1.07–1.31	0.001	1.05	0.94–1.16	0.406
Obesity	39.9%	1.10	0.97–1.25	0.158	1.00	0.88–1.15	0.953
**Subjective body weight estimation**							
Normal weight	39.9%	1			1		
Overweight, trying to reduce weight	37.7%	0.91	0.82–1.01	0.078	0.97	0.87–1.08	0.526
Overweight, not trying to reduce weight	49.1%	1.45	1.14–1.86	0.003	1.47	1.14–1.88	0.003
Overweight, feeling comfortable	37.7%	0.91	0.75–1.11	0.365	0.92	0.75–1.13	0.431

Results are based on logistic regression analysis. Data are presented as odds ratios (OR) with a 95% confidence interval (95% CI).

## Data Availability

The data presented in this study are available on request from the corresponding author.

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
