# Peer review of "Obesity, Fruit and Vegetable Intake, and Physical Activity Patterns in Austrian Farmers Compared to the General Population"

_ijerph, 2022, doi:10.3390/ijerph19159194_

Round 1

Reviewer 1 Report

Congratulations for very well written article. An average but correctly written study design. Measurement of motor activity by opinion is otherwise justified, but less reliable. Otherwise interesting data that has well drawn conclusions.

Author Response

Attached, you can find our responses.

Reviewer 2 Report

This article, based on secondary analysis of national level survey data, is well put together and careful in its claims. I believe it’s close to publishable (admit I didn’t check any of the stats), but I still have a couple of comments to offer in the spirit of helping improve it.

 The only really major question – on lines 329-332, the authours sensibly suggest that with the self-reported nature of the study on farmers, and the fact it was for insurance purposes, that alcohol intake might be under reported. That’s true. But would it possibly also be equally true that other things this paper is concerned with – weight, physical activity, for example, might be under or over-reported in order to make them seem healthier (and less of an insurance problem)? Suggest that there should be a section, either in the discussion, or the description in the methods, that deals with this issue. You work with what you can, but this survey could, theoretically, be quite off when compared to reality in lots of ways. I’m not saying it invalidates using it, but that there should be some discussion of the issue more broadly than just that one mention near the end of the paper.  

 When it comes to health promotion, I’m often skeptical of claims that “it’s a matter of knowledge” especially when it comes to things like nutrition and physical activity. Knowledge seems to be what we easily fall back on, because it’s much easier than other interventions. Knowledge gets a brief mention on lines 361-362. But in this case, I think more could be made questions of knowledge, especially with a link to reference #42. If that study was a good one, and it found that knowledge was a dominant factor in “rural population,” then maybe a little more forceful suggestion for health promotion of knowledge could be made here.  

 In lines 107- 100, the question on fruit servings and alcohol servings really confused me. Was that the way the study question was phrased, or is that the authours’ re-write? If it’s a recombination, I’d suggest separating the fruit and alcohol questions for clarity. If this is indeed how the survey question was phrased, then perhaps this requires some reflection from the authours. I read it as being confusing, in that maybe some people would interpret it not as two questions. Maybe this is why the study results find relatively low alcohol consumption in the farm population – some confusion with the intent of the original question?

 More minor things:

 Could we separate out the two parts of the second paragraph of the results? It seems like there’s an attempt to link education levels and health, even though it’s just the descriptive part.

 Line 94 – suggest the word “assembled” is changed to “combined,” just for clarity.

Author Response

Attached, you can find the responses.

Reviewer 3 Report

Dear Authors, 

Thank you so much for submitting this interesting manuscript. 

This is an interesting topic and there is not much literature available, especially from Europe. 

Abstract:

Well-written and contain all important information. The only aspect missing and if the word count allows, it would be useful to have the male: female ratio of the participants.

Introduction:

You can also add that the majority of the smallholder farmers are the ones going hungry in developing countries.

https://www.fao.org/neareast/news/view/en/c/1197345/

Methods:

You asked the question about physical activity but wouldn’t some of the farming be considered an active job?

Author Response

Attached, you can find our responses.

Reviewer 4 Report

There is a major flaw in your research design. You cannot compare self-report to interviewer assisted self report for dietary analysis. People are not always aware that they are eating vegetables, particularly if someone else is doing the food preparation. There is also a lack of knowledge as to what constitutes a vegetable. A meal of chicken, rice, and beans would include vegetables since beans can be counted as a vegetable when there is another protein source. Another example of lack of awareness fo what constitutes a vegetable is coleslaw. It is raw for people not to eat any vegetables. Although your comparisons are statistically different, they are not clinically different. Most people regardless of whether farmers or not did not consume 5 or more vegetables per day. The scale of zero, 1-4, 5 or more does not differentiate between people who are almost meeting the requirements (3-4 servings) from those with less than 1 or 1-2 servings. Again, these two methods cannot be compared do to the vast difference in how the data was collected (trained interviewers or no interviewers).

Author Response

Attached, you can find our responses.

Round 2

Reviewer 4 Report

Although you have clarified the methods, the study design is still flawed. You cannot compare interview with self-report for dietary data.

The metric for healthy eating is vegetables, yet only fruit is significantly different between farmer and the general population. 
